# Critic Regularized Regression

**Ziyu Wang**[*][†]
ziyu@google.com

**Alexander Novikov**[*]
anovikov@google.com

**Konrad Żołna**[*]
kondiz@google.com

**Jost Tobias Springenberg**[*]
springenberg@google.com

**Scott Reed**[*]
reedscot@google.com

**Bobak Shahriari**[*]
bshahr@google.com

**Noah Siegel**[*]
siegeln@google.com

**Josh Merel**[*]
jsmerel@google.com

**Caglar Gulcehre**[*]
caglarg@google.com

**Nicolas Heess**[*]
heess@google.com

**Nando de Freitas**[*]
nando@google.com

## Abstract

Offline reinforcement learning (RL), also known as batch RL, offers the prospect of policy optimization from large pre-recorded datasets without online environment interaction. It addresses challenges with regard to the cost of data collection and safety, both of which are particularly pertinent to real-world applications of RL. Unfortunately, most off-policy algorithms perform poorly when learning from a fixed dataset. In this paper, we propose a novel offline RL algorithm to learn policies from data using a form of critic-regularized regression (CRR). We find that CRR performs surprisingly well and scales to tasks with high-dimensional state and action spaces – outperforming several state-of-the-art offline RL algorithms by a significant margin on a wide range of benchmark tasks.

## 1 Introduction

Deep reinforcement learning (RL) algorithms have succeeded in a number of challenging domains. However, few of these domains have involved real-world decision making. One important reason is that online execution of policies during learning, which we refer to as *online RL*, is often not feasible or desirable because of cost, safety and ethics [8]. This is clearly the case in healthcare, industrial control and robotics. Nevertheless, for many of these domains, large amounts of historical data are available. This has led to a resurgence of interest in *offline RL* methods, also known as batch RL [19], which aim to learn policies from logged data without further interaction with the real system.

This interest in offline RL is further amplified by the evaluation crisis in RL: RL involves a close interplay between exploration and learning from experiences, which makes it difficult to compare algorithms. By decoupling these two problems and focusing on learning from fixed experiences, it becomes possible to share datasets with benchmarks to improve collaboration and evaluation in the field.

The naive application of off-policy RL algorithms with function approximation to the offline setting has often failed, and has prompted several alternative solutions [e.g. 9, 17, 30]. A shared conclusion is that failure stems from overly optimistic Q-estimates, as well as inappropriate extrapolation beyond

---

[*]DeepMind, London, United Kingdom.
[†]Google Brain, Toronto, Canada.

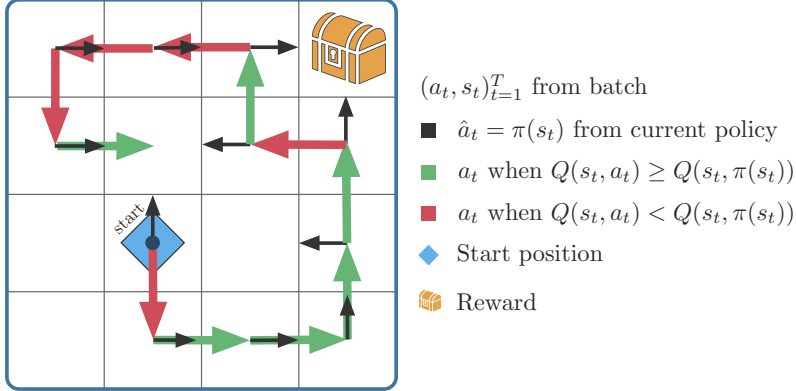

Figure 1: Illustration of the main idea behind CRR. The task is to reach the reward from the starting position as fast as possible. Consider learning a policy from the suboptimal (red/green) trajectory. For every state $s_t$, the action proposed by the current (suboptimal) policy $\pi(s_t)$ is shown with black arrows. CRR compares the critic prediction of the value $Q(s_t, a_t)$ of the action $a_t$ from the trajectory against the value $Q(s_t, \pi(s_t))$ of the action from the policy $\pi$. If $Q(s_t, a_t) \geq Q(s_t, \pi(s_t))$, the corresponding action is marked green and the pair $(s_t, a_t)$ is used to train the policy. If $Q(s_t, a_t) < Q(s_t, \pi(s_t))$, the action is marked red and is not used to train the policy. Thus, CRR filters out bad actions and enables learning better policies from low-quality data.

the observed data. This is especially problematic in combination with bootstrapping, where the Q-function is queried for actions that were not observed, and where errors can accumulate.

In this paper, we propose a novel offline RL algorithm to learn policies from data using a form of *critic-regularized regression* (CRR). CRR essentially reduces offline policy optimization to a form of value-filtered regression that requires minimal algorithmic changes to standard actor-critic methods. It is therefore easy to understand and implement. Figure 1 provides an intuitive explanation of CRR.

Despite the apparent simplicity of CRR, our experiments show that it outperforms several state-of-the-art offline RL algorithms by a significant margin on a wide range of benchmark tasks. Moreover, and very importantly, it scales to tasks with high-dimensional state and action spaces, and does well on datasets of diverse or low-quality data.

## 2   Related Work

For comprehensive in-depth reviews of offline RL, we refer the reader to Lange et al. [19] and Levine et al. [20]. The latter provides and extensive and very recent appraisal of the field.

*Behavior cloning* (BC) [28] is the simplest form of offline learning. Starting from a dataset of state-action pairs, a policy is trained to map states to actions via a supervised loss. This approach can be surprisingly effective when the dataset contains high-quality data, e.g. trajectories generated by an expert for the task of interest; see Merel et al. [22] for a large scale application. However, it can easily fail (i) when the dataset contains a large proportion of random or otherwise task irrelevant behavior; or (ii) when the learned policy induces a trajectory distribution that deviates far from that of the dataset under consideration [29].

Off-policy deep RL algorithms provide an alternative to BC. Unlike BC, these methods can take advantage of reward functions to outperform the demonstrator, see e.g. Cabi et al. [6]. Recently, it was shown that distributional off-policy deep RL techniques [5, 4], unlike their non-distributional counterparts, work well for offline RL in Atari [2] and robot manipulation [6]. A recent paper by Fujimoto et al. [9] has corroborated that distributional variants by themselves can be more effective, but under-perform in comparison to an algorithm purposely designed for offline RL, known as Batch-Constrained deep Q-learning (BCQ) [10].

Several methods have been proposed to overcome problems with off-policy deep RL in the offline setting. These failures have mostly been attributed to inappropriate generalization and overly confident Q estimates. One line of work focused on constraining the action choices to the support of the training data [10, 17, 30, 15]. This can be achieved by first learning a generative model of the data, and then sampling actions from this model for Q-learning [10]. The generative model can also be used with

constrained optimization to restrict the estimated RL policy [17, 15]. Both Fujimoto et al. [10] and Kumar et al. [17] further employ multiple Q-function estimates to reduce the optimism bias.

A second line of attack in offline RL has been to perform weighted behavior cloning. The idea is to use an estimate of the advantage function to select the best actions in the dataset for BC. This is similar in spirit to the motivation provided in Figure 1. Examples of this approach include monotonic advantage re-weighted imitation learning (MARWIL) [37], best-action imitation learning (BAIL) [7], advantage-weighted behavior models (ABM) [30] and advantage weighted regression [27], which has previously been studied in the form of a Fitted Q-iteration algorithm with low-dimensional policy classes [26]. Our approach differs from most of these in that we do not rely on observed returns for advantage estimation – as elaborated on in Sec. 3. Additionally, we introduce a technique we call Critic Weighted Policy (CWP) that uses the learned critic to improve results at test time.

A simple (binary) version of the filtering component of CRR was proposed by Nair et al. [25], whose results showed that the filter simply accelerated training, but gave mixed results in long horizons. An earlier version of a similar algorithm can be found in van Hasselt and Wiering [36]. It is important to note that these works perform filtered BC but in online settings, in which further data collection is allowed, and hence filtering may play a lesser role. The online setting is considerably more forgiving than the offline setting, which is the focus of this paper. Indeed, our analysis in Section 4 demonstrates that in the offline setting, design elements such as sample estimates for binary and exponential filtering, policy improvement with CWP, carefully designed deep recurrent networks, and the use of distributional value functions make a dramatic difference in the quality of the results.

# 3 Critic Regularized Regression

We derive Critic Regularized Regression (CRR), a simple, yet effective, method for offline RL.

## 3.1 Policy Evaluation

Suppose we are given a fixed dataset $\mathcal{B}$ containing either individual transition tuples $(\mathbf{s}_t, \mathbf{a}_t, r_t, \mathbf{s}_{t+1})$ or $K$-step trajectories $(\mathbf{s}_t, \mathbf{a}_t, r_t, \mathbf{s}_{t+1}, \mathbf{a}_{t+1}, \cdots, \mathbf{s}_{t+K})$. For brevity we will consider only the first case, but all algorithmic developments directly apply for $K$-step trajectories. An important step in many off-policy learning approaches is off-policy policy evaluation. This often involves learning an approximation to the Q-function minimizing a temporal difference loss similar to the following

$$\mathbb{E}_{\mathcal{B}}\Big[D\big(Q_\theta(\mathbf{s}_t, \mathbf{a}_t), (r_t + \gamma \mathbb{E}_{\mathbf{a} \sim \pi(\mathbf{s}_{t+1})} Q_{\theta'}(\mathbf{s}_{t+1}, \mathbf{a})))\big)\Big], \tag{1}$$

where the expectation over data is approximated by sampling $(\mathbf{s}_t, \mathbf{a}_t, r_t, \mathbf{s}_{t+1}) \sim \mathcal{B}$ and $D$ is some divergence measure; measuring discrepancy between the current estimate of the action-value $Q_\theta(\mathbf{s}_t, \mathbf{a}_t)$ (for policy $\pi$) and the td-update, based on a target network [24]. We note that we make use of a distributional Q-function as in Barth-Maron et al. [4], instead of using a squared error for $D$.

As discussed in recent work, [e.g. 10, 17, 30], without environment interaction (the offline RL setting) this loss may be problematic due to the "bootstrapping" issue mentioned above; where we evaluate the value of the next state $\mathbf{s}_{t+1}$ by $\mathbb{E}_{\mathbf{a} \sim \pi(\mathbf{s}_{t+1})} Q_{\theta'}(\mathbf{s}_{t+1}, \mathbf{a})$. If trained with standard RL, a parametric $\pi(\cdot|\mathbf{s}_{t+1})$ is likely to extrapolate beyond the training data and to propose actions that are not contained in $\mathcal{B}$. For these actions $Q$ will not have been trained and may produce bad estimates.

## 3.2 Policy Learning with CRR

To mitigate the aforementioned problem, we want to avoid evaluating $Q$ for $(\mathbf{s}, \mathbf{a}) \notin \mathcal{B}$. Thus, we aim to train $\pi$ by discouraging it from taking actions that are outside the training distribution. Such a requirement would be hard to achieve with standard policy gradients [e.g 31, 12, 33]. We thus change our objective to match the state-action mapping contained in the training data – but filtered by the Q-function. Specifically, we optimize

$$\arg\max_\pi \mathbb{E}_{(\mathbf{s}, \mathbf{a}) \sim \mathcal{B}}\Big[f(Q_\theta, \pi, \mathbf{s}, \mathbf{a}) \log \pi(\mathbf{a}|\mathbf{s})\Big], \tag{2}$$

where $f$ is a non-negative, scalar, function whose value is monotonically increasing in $Q_\theta$. Fundamentally, Eq. (2) tries to copy actions that exist in the data, thereby restricting $\pi$. Further properties

---

**Algorithm 1:** Critic Regularized Regression

---

**Input:** Dataset $\mathcal{B}$, critic net $Q_\theta$, actor net $\pi_\phi$, target actor and critic nets: $\pi_{\phi'}$, $Q_{\theta'}$, function $f$

**for** $n_{updates}$ **do**

    Sample $(\mathbf{s}_t^i, \mathbf{a}_t^i, r_t^i, \mathbf{s}_{t+1}^i)_{i=1}^b$ from $\mathcal{B}$.

    Update actor (policy) with gradient: $\nabla_\phi - \frac{1}{b} \sum_i \log \pi_\phi(\mathbf{a}_t^i | \mathbf{s}_t^i) f(Q_\theta, \pi_\phi, \mathbf{s}_t^i, \mathbf{a}_t^i)$

    Update critic with gradient: $\nabla_\theta - \frac{1}{b} \sum_i D\left[Q_\theta(\mathbf{s}_t^i, \mathbf{a}_t^i), (r_t^i + \gamma \mathbb{E}_{\mathbf{a} \sim \pi_{\phi'}(\mathbf{s}_{t+1}^i)} Q_{\theta'}(\mathbf{s}_{t+1}^i, \mathbf{a}))\right]$

    Update the target actor/critic nets every $N$ steps by copying parameters: $\theta' \leftarrow \theta, \ \phi' \leftarrow \phi$.

**end**

---

of the learning objective can be controlled through different choices of $f$, for example when $f := 1$, Eq. (2) is equivalent to Behavioral Cloning (BC). The success of BC is, however, highly dependent on the quality of the dataset $\mathcal{B}$. When $\mathcal{B}$ does not contain enough transitions generated by a policy performing well on the task, or the fraction of poor data is too large, then BC is likely to fail.

Provided $Q$ is sufficiently accurate for $(\mathbf{s}, \mathbf{a}) \in \mathcal{B}$ (e.g. learned using Eq. (1)), we can consider additional choices of $f$ that enable off-policy learning to overcome this problem:

$$f := \mathbb{1}[\hat{A}_\theta(\mathbf{s}, \mathbf{a}) > 0], \tag{3}$$

$$f := \exp\left(\hat{A}_\theta(\mathbf{s}, \mathbf{a})/\beta\right), \tag{4}$$

where $\beta > 0$ is a hyper-parameter, $\mathbb{1}$ the indicator function, and $\hat{A}_\theta(\mathbf{s}, \mathbf{a})$ is an estimated advantage function. Eq. (2) bears similarity to the objective for training the behavior prior in [30]. More precisely, by using $f$ from Eq. (3) and $\hat{A}_K$ (see Table 1) we recover the form of the ABM 'prior-policy'. Intuitively, Eq. (3) entails BC on a filtered dataset where the filtering increases the average quality of the actions we learn from. The filter is defined in terms of the value of the current policy.

**Exponential weighting as regularized policy iteration.** We can gain insight into Eq. (4) by realizing that it approximately implements a regularized policy improvement step in a policy iteration scheme. Consider

$$\mathcal{L}(q, \mu_\mathcal{B}) = \mathbb{E}_{\mathbf{s} \sim \mathcal{B}}\left[\mathbb{E}_q[Q_\theta(\mathbf{s}, \mathbf{a})] - \beta \mathrm{KL}[q(\cdot|\mathbf{s}), \mu_\mathcal{B}(\cdot|\mathbf{s})]\right], \tag{5}$$

where $\mu_\mathcal{B}$ is the policy that generated the data. This objective is similar to the off-policy improvement from MPO [1] – but with $\mu_\mathcal{B}$ instead of the policy prior. The optimal policy for this objective can be written as $q(\mathbf{a}|\mathbf{s}) = \exp(Q_\theta(\mathbf{s},\mathbf{a})/\beta)\mu_\mathcal{B}(\mathbf{a}|\mathbf{s})/Z(s)$. $Z(s)$ is hard to evaluate in our setting (we only have access to $\mathcal{B}$) so we use $q(\mathbf{a}|\mathbf{s}) \propto \exp(\hat{A}_\theta(\mathbf{s}, \mathbf{a})/\beta)$ in practice, thus performing an implicit, approximate per state normalization by subtracting $V(s)$. Projecting this distribution back onto the parametric policy $\pi_\phi$ can be done by minimizing the cross-entropy $H[q(\cdot|\mathbf{s}), \pi_\phi(\cdot|\mathbf{s})]$. Again, to avoid computing the normalizing constant for $q$, we use samples from the dataset $\mathcal{B}$ to estimate this cross-entropy, which yields Eq. (2) with $f$ chosen as in Eq. (4). Minimizing this cross-entropy corresponds to "sharpening" the action distribution $\mu_\mathcal{B}$, giving higher weight to better actions. Further, as $\beta \to \infty$ this objective becomes behavior cloning (BC).

**Theoretical analysis in the tabular setting.** In the Appendix, we show that in the tabular setting, CRR is safe and improves upon the behavioral policy defined by the dataset.

**On the choice of advantage estimators.** Since we are interested in high-dimensional action spaces, we use a sample estimate of the advantage in CRR (Algorithm 1). We contrast different choices for $\hat{A}$ in Table 1. From these we consider $\hat{A}_{\mathrm{mean}}$ but found that for small $m$ it may overestimate the advantage due to stochasticity. For $f$ as in Eq. (3) this could lead to sub-optimal actions being included. We therefore also consider $\hat{A}_{\mathrm{max}}$; a pessimistic estimate of the advantage.

Except for the work on FQI-AWR [26] – where the authors considered a simpler setting, learning both Q and V with simple kernel based regression followed by policy learning based on advantage weighting – the recent literature has mainly considered K-step or Monte-Carlo return based estimates of the advantage ($\hat{A}_K$, $\hat{A}_{MC}$ in the table). Among these, our formulation in Eq. (3) and Eq. (4) is similar to [30] as highlighted above. It also bears some similarity to recent work on advantage weighted regression [37, 27], which uses $\hat{A}_K$, or $\hat{A}_{MC}$ with a learned $V$-function to form advantage estimates. A choice that we find to not work well in many offline RL scenarios, as we explain below.

Table 1: Advantage estimates considered by different algorithms.

| Advantage estimate | Algorithms |
|---|---|
| $\hat{A}_{\text{mean}}(\mathbf{s}_t, \mathbf{a}_t) = Q_\theta(\mathbf{s}_t, \mathbf{a}_t) - \frac{1}{m}\sum_{j=1}^{m} Q_\theta(\mathbf{s}_t, \mathbf{a}^j)$, with $\mathbf{a}^j \sim \pi(\cdot|\mathbf{s}_t)$ | CRR,FQI-AWR[26] |
| $\hat{A}_{\text{max}}(\mathbf{s}_t, \mathbf{a}_t) = Q_\theta(\mathbf{s}_t, \mathbf{a}_t) - \max_{j=1}^{m} Q_\theta(\mathbf{s}_t, \mathbf{a}^j)$, with $\mathbf{a}^j \sim \pi(\cdot|\mathbf{s}_t)$ | CRR(max) |
| $\hat{A}_k(\mathbf{s}_t, \mathbf{a}_t) = \gamma^K V(\mathbf{s}_{t+K}) + \sum_{t'=t}^{t+K-1} \gamma^{t'-t} r(\mathbf{s}_{t'}, \mathbf{a}_{t'}) - V(\mathbf{s}_t)$ | ABM[30] |
| $\hat{A}_{MC}(\mathbf{s}_t, \mathbf{a}_t) = \sum_{t'=t}^{\infty} \gamma^{t'-t} r(\mathbf{s}_{t'}, \mathbf{a}_{t'}) - V(\mathbf{s}_t)$ | MARWIL[37],AWR[27] |

A full listing of our algorithm is given in 1; where we also adopt standard practices from the deep RL literature to stabilize training (e.g. target networks) and note that we use a distributional critic.

**CRR vs. return-based methods.** Using K-step returns (with K > 1) can be problematic when the data is very off-policy as we will show experimentally in the appendix. Additionally, in stochastic environments, using K-step (or episodic) returns for estimating advantages may lead to risk seeking or otherwise undesirable behavior. To illustrate this point, let us consider the following simple two-armed bandit problem: the first arm generates a payoff of either $0$, or $1$ with $50\%$ probability each. The second arm generates a deterministic payoff of $0.9$ – it hence should be the favored arm. Let us assume that $2/3$ of the actions in our data-set pull the first arm and the remaining $1/3$ the second arm. A perfectly learned $Q$ function would clearly favor arm two over arm one. By using returns $R$ for advantage estimation (computing advantages as $R - V$), however, Advantage weighted regression (AWR) [27], or similar methods, would favor the first arm over the second arm; regardless of the temperature parameter. Similarly, the prior learned in ABM [30] would choose the two actions equally and therefore also fall short. For ABM, the additional RL step could potentially recover the correct action choice (depending on how far the RL policy is allowed to deviate from the prior) but no guarantee exists that it will. Last but not least, methods that rely on reward filtering would also favor arm one over arm two [7, 18].

**Critic Weighted Policy (CWP).** The learned $Q$-function can be used during policy execution to perform an additional approximate policy improvement step. Indeed, search-based methods have made use of this fact in discrete domains [32]. We can consider the solution of Eqn. (5) albeit now using the learned policy $\pi_\phi$ as the prior. The solution is given by $\bar{q} = \arg\max_{\bar{q}} \mathcal{L}(\bar{q}, \pi_\phi)$, which yields $\bar{q}(\mathbf{a}|\mathbf{s}) = \exp(Q_\theta(\mathbf{s},\mathbf{a})/\beta)\pi_\phi(\mathbf{a}|\mathbf{s})/Z(s)$. We can use this policy instead of $\pi$ during action selection. To sample from $\bar{q}$, we use importance sampling. We first sample actions $\mathbf{a}_{1:n}$ from $\pi_\phi(\cdot|\mathbf{s})$, weight the different actions by their importance weights $\exp(Q_\theta(\mathbf{s}, \mathbf{a}_i)/\beta)$ and finally choose an action by re-sampling with probabilities $P(\mathbf{a}_i) = \exp(Q_\theta(\mathbf{s},\mathbf{a}_i)/\beta)/\sum_{j=1}^{n} \exp(Q_\theta(\mathbf{s},\mathbf{a}_j)/\beta)$. Note that this corresponds to self-normalized importance sampling and does not require access to $Z(s)$.

## 4  Experiments

We evaluate our algorithm, CRR, on a number of challenging simulated manipulation and locomotion domains. Several of our tasks involve very high-dimensional action spaces as well as perception via RGB cameras (subject to weak partial observability due to egocentricity in a locomoting body). Our

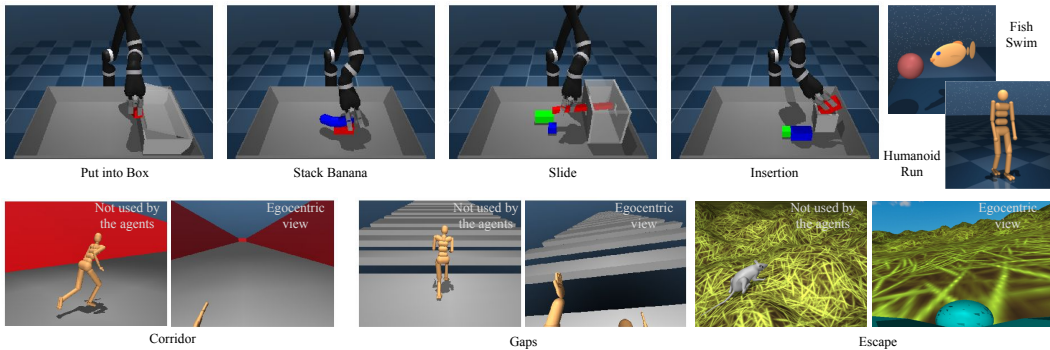

Figure 2: Illustrations of the environments. The agent has to solve the manipulation tasks using vision provided via 2 cameras. In the locomotion domains, a head-mounted egocentric camera is used to sense the surroundings. In the Deepmind control suite domains, no visual input is needed.

Table 2: Results on Deepmind Control suite. We divide the Deepmind control suite environments into two rough categories: easy (first 6) and hard (last 3).

| | BC | D4PG | ABM | BCQ | CRR exp | CRR binary | CRR binary max |
|---|---|---|---|---|---|---|---|
| Cartpole Swingup | $386 \pm 6$ | $855 \pm 13$ | $798 \pm 30$ | $444 \pm 15$ | $664 \pm 22$ | $\mathbf{860 \pm 7}$ | $858 \pm 15$ |
| Finger Turn Hard | $261 \pm 39$ | $764 \pm 24$ | $566 \pm 25$ | $311 \pm 38$ | $714 \pm 38$ | $755 \pm 31$ | $\mathbf{833 \pm 57}$ |
| Walker Stand | $386 \pm 6$ | $\mathbf{929 \pm 46}$ | $689 \pm 13$ | $501 \pm 5$ | $797 \pm 30$ | $881 \pm 13$ | $\mathbf{929 \pm 10}$ |
| Walker Walk | $417 \pm 33$ | $939 \pm 19$ | $846 \pm 15$ | $748 \pm 24$ | $901 \pm 12$ | $936 \pm 3$ | $\mathbf{951 \pm 7}$ |
| Cheetah Run | $407 \pm 56$ | $308 \pm 121$ | $304 \pm 32$ | $368 \pm 129$ | $\mathbf{577 \pm 79}$ | $453 \pm 20$ | $415 \pm 26$ |
| Fish Swim | $466 \pm 8$ | $281 \pm 77$ | $527 \pm 19$ | $473 \pm 36$ | $517 \pm 21$ | $585 \pm 23$ | $\mathbf{596 \pm 11}$ |
| Manipulator Insert Ball | $385 \pm 12$ | $154 \pm 54$ | $409 \pm 4$ | $98 \pm 29$ | $625 \pm 24$ | $\mathbf{654 \pm 42}$ | $636 \pm 43$ |
| Manipulator Insert Peg | $324 \pm 31$ | $71 \pm 2$ | $345 \pm 12$ | $194 \pm 117$ | $\mathbf{387 \pm 36}$ | $365 \pm 28$ | $328 \pm 24$ |
| Humanoid Run | $382 \pm 2$ | $1 \pm 1$ | $302 \pm 6$ | $22 \pm 3$ | $\mathbf{586 \pm 6}$ | $412 \pm 10$ | $226 \pm 11$ |

results demonstrate that CRR works well even in these challenging settings and that it outperforms previously published approaches, in some cases by a considerable margin. We also perform several ablations that highlight the importance of individual algorithm components, and finally provide results on some toy domains that provide insight on why alternative approaches may fail.

## 4.1 Environments and datasets

We experiment with the continuous control tasks introduced in RL Unplugged (RLU) [3]. There are 17 different tasks in RLU: nine tasks from the Deepmind Control suite [34] and seven locomotion tasks. We additionally introduce four robotic manipulation datasets. The tasks cover a diverse set of scenarios, making our experimental study one of the most comprehensive to date for offline RL. All simulations are conducted using MuJoCo [35]; illustrations of the environments are given in Fig. 2.

**Deepmind Control Suite (DCS).** We consider the following tasks: *cartpole-swingup*, *walker-stand*, *walker-walk*, *cheetah-run*, *finger-turn-hard*, *manipulator-insert-ball*, *manipulator-insert-peg*, *fish-swim*, *humanoid-run*, all from the DeepMind Control Suite. The data contains both successful and unsuccessful episodes. Many of these tasks are relatively straightforward with low action dimensions. Observations are given by features (no pixel observations nor partial observability). A number of these tasks are, however, quite challenging. Especially for the high-dimensional humanoid body, generating the dataset from several independent learning experiments lead to very diverse data; posing challenges for offline RL algorithms. For a split between easy and hard tasks, please see Table 2.

**Locomotion.** We further consider challenging locomotion tasks from RLU, for a humanoid [adapted from 13, 21] as well as for a rodent [adapted from 23]. The three humanoid tasks require running down the corridor at a target speed (*corridor* task), avoiding obstacles like walls (*walls* task) or gaps (*gaps* task). Data for these tasks is generated by training a hierarchical architecture that uses a pre-trained low-level controller (NPMP), following Merel et al. [22]. Note that we use the pre-trained controller only for generating the data sets but *not* in our offline experiments. For the rodent, there are four tasks comprising "escaping" from a hilly region (*escape* task), foraging in a maze (*forage* task), an interval timing task (*two-tap* task), and a size-proportionate version of the gaps task (*gaps* task) [for details, see 23]. As before, 3 independent online agent training runs are used to record the data for each of the tasks. The data is again of varying quality as it includes failed trajectories from early in training. This set of tasks is particularly challenging due to their high dimensional action spaces (56DoF for humanoid and 38DoF for the rodent). Additionally, the agent must observe the surroundings, to avoid obstacles, using an unstable egocentric camera (controlled via head and neck movements). Last but not least, a few tasks in this task suite are partially observable and thus require recurrent agents.

**Robotic Manipulation.** These tasks require the agent to control a simulated Kinova Jaco robotic arm (9DoF) to solve a number of manipulation problems. We use joint velocity control (at 20HZ) of all 6 arm joints and the 3 joints of the hand. The agent observes the proprioceptive features directly, but can only infer the objects on the table from pixel observations. Two camera views of size $64 \times 64$ are provided: one frontal camera covering the whole scene, and an in-hand camera for closeup of the objects. The episodes are of length 400 and the reward function is binary depending on whether the task is successfully executed. We consider 4 different challenges: *put into box*, *stack banana*,

Table 3: Results on Locomotion Suite. The first 3 tasks can be solved by feedforward agents; the corresponding datasets are not sequential. The last 4 tasks necessitate observation histories and all agents here are recurrent.

|  | BC | D4PG | ABM | CRR exp | CRR binary | CRR binary max |
|---|---|---|---|---|---|---|
| Humanoid Corridor | $220 \pm 194$ | $4 \pm 4$ | $64 \pm 3$ | $\mathbf{918 \pm 14}$ | $484 \pm 97$ | $245 \pm 180$ |
| Humanoid Gaps | $149 \pm 9$ | $5 \pm 3$ | $94 \pm 9$ | $\mathbf{546 \pm 36}$ | $149 \pm 89$ | $33 \pm 21$ |
| Rodent Gaps | $463 \pm 137$ | $176 \pm 6$ | $420 \pm 70$ | $\mathbf{957 \pm 19}$ | $492 \pm 117$ | $392 \pm 6$ |
| Humanoid Walls | $138 \pm 77$ | $2 \pm 1$ | $131 \pm 25$ | $\mathbf{422 \pm 24}$ | $289 \pm 72$ | $232 \pm 24$ |
| Rodent Escape | $388 \pm 3$ | $24 \pm 14$ | $441 \pm 16$ | $428 \pm 26$ | $444 \pm 45$ | $\mathbf{499 \pm 40}$ |
| Rodent Mazes | $343 \pm 48$ | $53 \pm 1$ | $\mathbf{478 \pm 7}$ | $459 \pm 7$ | $464 \pm 12$ | $457 \pm 13$ |
| Rodent Two Tap | $325 \pm 60$ | $16 \pm 2$ | $598 \pm 2$ | $543 \pm 32$ | $\mathbf{615 \pm 19}$ | $588 \pm 14$ |

*slide* and *insertion*. The dataset for each task is generated from 3 independent runs of a DPGfD agent (8000 episodes each).[3]

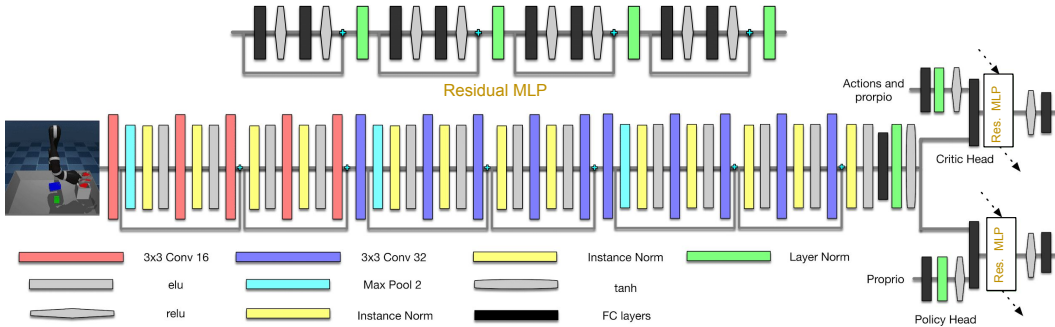

Figure 3: [TOP] Residual MLP used as a part of our network architecture. The black blocks indicated linear layers (of size 1024), green blocks layer-norm, and gray blocks ReLUs. [BOTTOM] Full CRR network. The residual MLP fits on top of the CRR networks. If recurrence is needed, we add two LSTM layers of size 1024 on top of the residual MLP layers before producing value and policy.

## 4.2 Experimental Setup

For environments where vision is involved we use ResNets to process the visual inputs. Proprioceptive information is concatenated with the output of the ResNets and the result is fed into a MLP with residual connections. The network structures are depicted in Fig. 3. The critic and policy networks share the vision modules and maintain separate copies of MLPs of identical structure, but employ different last layers to compute the $Q$ and policy respectively. In our experiments, we use 4 residual blocks for the MLPs. For environments where vision is not required, we use the MLP alone. For CRR policies, we use a mixture of Gaussians policy head with 5 mixture components. Crucially, when evaluating CRR (and BC) policies, we turn off the noise in the Gaussian component distributions. We show this is essential in achieving good results in supplementary materials. We also provide additional details on our evaluation protocol in the appendix. For D4PG we compared using the same architecture as for CRR versus using the architecture from [14] and found the latter to be superior, so we used it for all experiments. The results are presented in Table 2, 3, and Figure 4.

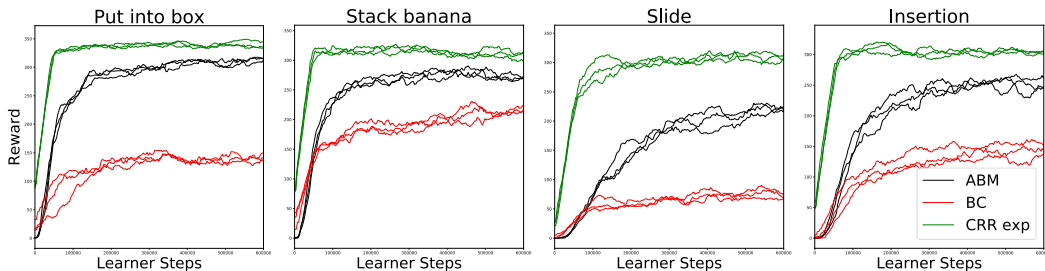

Figure 4: Results for CRR, ABM and BC on manipulation datasets. Only offline data is used during training.

### 4.3 Analysis of CRR variants

In our experiments, we evaluate 3 variants of CRR which we call: *exp* (corresponding to Eqn. (4) with function $f$ clipped to the maximal value of 20 for stability), *binary* (Eqn. (3)), and *binary max* (Eqn. (3)). The first two variants use $\hat{A}_{\mathrm{mean}}$ and the last one uses $\hat{A}_{\mathrm{max}}$. Notice that all three variants do reasonably across all environments and their performance differ little in most environments.

When learning on low-complexity environments, however, CRR *binary* and especially CRR *binary max* are superior; see Table 2, row 1-4. We hypothesize that this is due to the fact that action filtering imposed by the *binary* (and more so for *binary max*) rule removes low-quality actions more aggressively. As the complexity of these environments is low, good policies can be learned from high quality data points, making CRR binary (max) particularly effective. In contrast, the *exp* rule tends to be too permissive and copies sub-optimal actions, leading to performance degradation.

Additionally, the relative value of an inferior action in the dataset compared to that of the policy tends to increase due to value overestimation. As training progresses, CRR therefore tends to copy more inferior actions as its policy improves. This could eventually lead to the decline of performance on some environments. In the appendix, we demonstrate the value overestimation phenomenon and introduce further analysis.

In contrast, for harder environments, where the task is difficult but high quality data is relatively abundant, CRR exp becomes superior. This may be explained by the fact that CRR binary filters out too many actions to adequately learn policies. For example, when training on the humanoid-run environment in DCS, CRR binary effectively considers, on average, $10\% - 12\%$ of actions in the entire dataset and CRR binary max $2\% - 5\%$. We speculate that this explains the superior performance of CRR exp on the humanoid tasks in DCS and the locomotion suites. On all manipulation environments the binary indicator performs similar to the exponential version.

CWP generally improves the performance of CRR. Consequently, for all CRR variants we present only results with CWP and leave the ablation to the appendix.

### 4.4 Comparison to baselines

**Comparison to D4PG.** We compare CRR with a state-of-the-art off-policy RL algorithm for continuous control: D4PG [4]. D4PG has been used successfully to solve a variety of problems [34] as well as for learning from a combination of offline (and off-task) and online data [6]. It utilizes the same distributional critic as CRR, allowing us to isolate the effect of changing the policy losses. We tuned the network architectures of D4PG on a few control suite tasks and found that it favors smaller networks compared to those used for CRR. The two approaches otherwise differ only in the policy training and how bootstrapping is performed during policy evaluation (for D4PG we query $Q(s, a)$ at the policy mean while CRR employs a sample based approximation). Full details on the hyper-parameters are given in the appendix. The results on manipulation as well as the DCS domains are shown in Fig. 4 and Tables 2 and 3.

Overall, our results confirm previously published results that the naive application of off-policy RL algorithms can fail in the offline RL settings [9, 30]: **D4PG** performs well on some of the DCS domains, but fails on the domains with higher dimensional action spaces (*humanoid-run*, and problems on the locomotion suite). Here its performance is indistinguishable from a random policy. This is also consistent with the results of [2] who find that standard off-policy RL algorithms can perform well in the offline setting in some cases; presumably when the state-action space of the domain is well covered in the dataset.

**Comparison to BCQ / ABM.** We also compare to two recently published offline RL algorithms: BCQ [10] and ABM [30]. We used the hyperparameters listed in [10] for BCQ, including network architectures and hyperparameters. To better ablate the difference between CRR and ABM, we implemented ABM to use exactly the same architectures as CRR with the exception of using Gaussian policy heads for ABM to stay close to the original MPO formulation [1]. Our implementation of ABM is thus equipped with distributional critics and can use recurrent networks where appropriate. We adopt hyper-parameters specific to ABM from the original paper. (see supplementary for details).

Similar to D4PG, BCQ behaves reasonably well on the easier tasks on DCS. It, however, fails to make progress on harder tasks like *humanoid-run* and *manipulator-insert-ball*. For that reason we do not consider BCQ for more complicated environments.

ABM performs well on most control suite environments. Compared to CRR, however, it underper-forms on the Humanoid Run task, as well as the humanoid tasks in the Locomotion Suite. These are the same tasks on which the performance of *CRR binary* is weaker, and we hypothesize that the lower performance can be explained by the structural similarity between *CRR bin* and the prior in *ABM* (see Section 3). Furthermore, the policy update in MPO is ineffective at reducing the stochasticity of the prior policy which can lower performance in humanoid and manipulation environments.

**Comparison to Behavior Cloning.** Although BC can work surprisingly well when high quality data for a task is available, its performance suffers in the presence of low-quality data. We use the same network architecture (without the critic network) and hyper-parameter as for CRR. This allows us to isolate the effect of the advantage-based filtering step that is part of CRR. As shown in the experiments, BC performs surprisingly well on a number of environments. Notably, unlike D4PG or BCQ, BC behaves reasonably well in environments with very high dimensional action spaces demonstrating its advantages over the RL losses. BC is, however, inferior to CRR or ABM overall.

## 5 Conclusion

We have presented an algorithm for offline RL that is simpler than existing methods but leads to surprisingly good performance even on challenging tasks. Our algorithm can be seen as a form of filtered behavioral cloning where data is selected based on information contained in the policy's Q-function. We have investigated several variants of the algorithm. CRR *exp* performs especially well across the entire range of tasks considered. Our detailed evaluation has highlighted that design factors, such as the choice of filter, can have significant influence depending on the nature of the task. We have provided some preliminary explanations of these effects. Given the already promising performance of CRR we believe that studying the underlying dynamics further is a valuable direction for future work and has the potential to reveal further algorithmic improvements that may push the frontier of offline RL algorithms in terms of robustness, performance and simplicity.

## Broader Impact

RL methods represent a unique solution principle that could lead to substantial progress in many real-world applications that are beneficial to society, such as the development of assistive robotic technologies for the disabled. As online RL and especially exploration is difficult (and sometimes dangerous) in the real world, offline RL provides a path for RL methods to be more broadly applied in practice. This paper introduces a new algorithm that could lead to improved performance on some real world tasks. As with all algorithms that can be used to automate decision making policies, however, offline RL methods could be used for applications with a negative impact on society. Since offline RL algorithms require existing datasets, we should also be vigilant when collecting datasets so as to avoid bias and prejudices.

## Footnotes

[3]Human demonstrations are used to train the DPGfD agents which additionally use distributional critics.

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
