[Supplementary Material]

# A Appendix

## A.1 Analysis of CRR in the tabular setting

In this section, we show that in the tabular setting, CRR is safe and improves upon the behavior policy defined by the dataset. In addition, we show that as the dataset grows, the policies learned by CRR perform sensibly in the true underlying environment.

We assume that there is an underlying Markov Decision Process (MDP) $(\mathcal{S}, \mathcal{A}, P_\mathcal{M}, r, \gamma)$. For simplicity, We assume finite state and action spaces and a deterministic reward function $r : \mathcal{S} \times \mathcal{A} \to \mathbb{R}$.

Consider a dataset of the format $\mathcal{B} = \{(\mathbf{s}_i, \mathbf{a}_i, \mathbf{s}_i')\}_i$. We assume the dataset is *coherent* (see [10]), meaning that if $(\mathbf{s}_i, \mathbf{a}_i, \mathbf{s}_i') \in \mathcal{B}$, then $(\mathbf{s}_i', \mathbf{a}_i', \mathbf{s}_i'') \in \mathcal{B}$ for some $\mathbf{a}_i', \mathbf{s}_i''$ unless $\mathbf{s}_i'$ is a terminal state.

Given the original MDP and data $\mathcal{B}$ we define the associated *empirical MDP* $M_\mathcal{B}$ as in Section 4.1 of [10]: The empirical MDP shares the same action space ($\mathcal{A}$), and state space ($\mathcal{S}$) along with an additional terminal state $\mathbf{s}_{term}$. $M_\mathcal{B}$ follows an empirical state transition probabilities:

$$P_\mathcal{B}(\mathbf{s}'|\mathbf{s}, \mathbf{a}) = \frac{N(\mathbf{s}, \mathbf{a}, \mathbf{s}')}{\sum_{\bar{\mathbf{s}}} N(\mathbf{s}, \mathbf{a}, \bar{\mathbf{s}})}$$

where $N(\mathbf{s}, \mathbf{a}, \mathbf{s}')$ is the count of the appearance of $\mathbf{s}, \mathbf{a}, \mathbf{s}'$ in the dataset. In the case where $\sum_{\bar{\mathbf{s}}} N(\mathbf{s}, \mathbf{a}, \bar{\mathbf{s}}) = 0$, we set $P_\mathcal{B}(\mathbf{s}_{term}|\mathbf{s}, \mathbf{a}) = 1$ and set $r(\mathbf{s}, \mathbf{a})$ to an arbitrary value. Assume that $(\mathbf{s}, \mathbf{a}) \sim \mathcal{B}$, we define an empirical policy distribution

$$\mu_\mathcal{B}(\mathbf{a}|\mathbf{s}) = \frac{N(\mathbf{a}, \mathbf{s})}{\sum_{\bar{\mathbf{a}}} N(\bar{\mathbf{a}}, \mathbf{s})},$$

where $N(\mathbf{a}, \mathbf{s}) = \sum_{\bar{\mathbf{s}}} N(\mathbf{a}, \mathbf{s}, \bar{\mathbf{s}})$. If $\sum_{\bar{\mathbf{a}}} N(\bar{\mathbf{a}}, \mathbf{s}) = 0$, then let $\mu_\mathcal{B}(\cdot|\mathbf{s})$ be uniform. We also define $d_\mathcal{B}(\mathbf{s}) = \frac{\sum_{\mathbf{a}, \mathbf{s}'} N(\mathbf{s}, \mathbf{a}, \mathbf{s}'))}{|\mathcal{B}|}$.

Given $M_\mathcal{B}$ and a policy $\pi$, we define the associated value functions

$$Q_\mathcal{B}^\pi(\mathbf{s}, \mathbf{a}) = \mathbb{E}_{\mathbf{s}_{t+i+1} \sim P_\mathcal{B}(\cdot|\mathbf{s}_{t+i}, \mathbf{a}_{t+i}), \mathbf{a}_{t+i} \sim \pi(\cdot|\mathbf{s}_{t+i})}\left[\sum_{i=0}^{\infty} \gamma^i r(\mathbf{s}_{t+i}, \mathbf{a}_{t+i}) \,\Big|\, \mathbf{s}_t = \mathbf{s}, \mathbf{a}_t = \mathbf{a}\right],$$

and $V_\mathcal{B}^\pi(\mathbf{s}) = \mathbb{E}_{\mathbf{a} \sim \pi(\cdot|\mathbf{s})} Q_\mathcal{B}^\pi(\mathbf{s}, \mathbf{a})$.

Given $Q_\mathcal{B}^{\pi_i}$, we consider the following CRR objectives in the tabular setting:

$$\text{Tab. CRR exp:} \quad \pi_{i+1} \quad \leftarrow \quad \arg\max_\pi \mathbb{E}_{\mathbf{s} \sim d_\mathcal{B}}\left[\sum_\mathbf{a} Q_\mathcal{B}^{\pi_i}(\mathbf{s}, \mathbf{a})\pi(\mathbf{a}|\mathbf{s})\right]$$

$$\text{subject to: } KL\Big(\pi(\cdot|\mathbf{s})||\mu_\mathcal{B}(\cdot|\mathbf{s})\Big) \leq \epsilon \; \forall \mathbf{s} \tag{6}$$

$$\text{Tab. CRR binary:} \quad \pi_{i+1} \quad \leftarrow \quad \arg\max_\pi \mathbb{E}_{\mathbf{s} \sim d_\mathcal{B}}\left[\sum_\mathbf{a} \mathbb{1}_{[Q_\mathcal{B}^{\pi_i}(\mathbf{s}, \mathbf{a}) \geq V^{\pi_i}(\mathbf{s})]} \mu_\mathcal{B}(\mathbf{a}|\mathbf{s}) \log \pi(\mathbf{a}|\mathbf{s})\right]. \tag{7}$$

Notice that the Tabular CRR exp objective looks different from the learning rule defined by Eqn. 4. However, it is easy to show that (using the KKT conditions)

$$\pi_{i+1}(\mathbf{a}|\mathbf{s}) = \frac{\exp\left(\frac{Q_\mathcal{B}^{\pi_i}(\mathbf{s}, \mathbf{a}) - V^{\pi_i}(\mathbf{s})}{\beta(\mathbf{s})}\right) \mu_\mathcal{B}(\mathbf{a}|\mathbf{s})}{Z^{\pi_{i+1}}(\mathbf{s})}, \tag{8}$$

where $\beta(\mathbf{s})$ is a state dependent factor and $Z^{\pi_{i+1}}(\mathbf{s})$ the normalization constant. This equation is in a much more familiar form albeit with a state dependent $\beta(\mathbf{s})$.

With the tabular CRR objectives defined, we introduce the CRR algorithms in the tabular setting as presented in Algorithm 2.

To illustrate why CRR is safe is the offline setting, we first show that policies trained via CRR (in the $M_\mathcal{B}$) would not try actions not present in the dataset.

---

**Algorithm 2:** Tabular Critic Regularized Regression

---

**Input:** Empirical MDP $M_{\mathcal{B}}$, and empirical policy $\mu_{\mathcal{B}}$
Start with $\pi_0 = \mu_{\mathcal{B}}$
**for** $i \in \{1, \cdots, \infty\}$ **do**
&emsp;| &emsp; Evaluate $Q_{\mathcal{B}}^{\pi_i}$ in the empirical MDP
&emsp;| &emsp; Compute $\pi_{i+1}$ according to Equation (6) or (7)
**end**

---

**Proposition 1.** supp $\pi_i(\cdot|\mathbf{s}) \subseteq$ supp $\mu_{\mathcal{B}}(\cdot|\mathbf{s})$ &emsp; $\forall i \geq 1, \mathbf{s} \in \mathcal{B}$ *for Tabular CRR exp (binary) objectives.*

*Proof.* We show the correctness of the statement only for the tabular CRR exp objective to avoid redundancy. Following Eqn. 8, we see that whenever $\mu_{\mathcal{B}}(\mathbf{a}|\mathbf{s}) = 0$, we have $\pi_{i+1}(\mathbf{a}|\mathbf{s}) = 0$. &emsp; □

In addition to being safe, we show that each iteration of CRR improves performance.

**Proposition 2.** *(Policy improvement) When using the Tabular CRR binary objective,* $Q_{\mathcal{B}}^{\pi_{i+1}}(\mathbf{s}, \mathbf{a}) \geq Q_{\mathcal{B}}^{\pi_i}(\mathbf{s}, \mathbf{a}) \, \forall \mathbf{s}, \mathbf{a} .$

*Proof.* Define

$$q_{i+1}(\mathbf{a}|\mathbf{s}) = \frac{\mathbb{1}_{[Q_{\mathcal{B}}^{\pi_i}(\mathbf{s}, \mathbf{a}) \geq V^{\pi_i}(\mathbf{s})]} \mu_{\mathcal{B}}(\mathbf{a}|\mathbf{s})}{Z^{\pi_{i+1}}(\mathbf{s})}$$

where $Z^{\pi_{i+1}}(\mathbf{s})$ is the partition function. Then it is easy to see that

$$\arg\max_{\pi} \mathbb{E}_{\mathbf{s} \sim d_{\mathcal{B}}} \left[ \sum_{\mathbf{a}} \mathbb{1}_{[Q_{\mathcal{B}}^{\pi_i}(\mathbf{s}, \mathbf{a}) \geq V^{\pi_i}(\mathbf{s})]} \mu_{\mathcal{B}}(\mathbf{a}|\mathbf{s}) \log \pi(\mathbf{a}|\mathbf{s}) \right]$$

$$= \arg\max_{\pi} \mathbb{E}_{\mathbf{s} \sim d_{\mathcal{B}}} \left[ KL\Big( q_{i+1}(\cdot|\mathbf{s}) || \pi(\cdot|\mathbf{s}) \Big) \right].$$

Therefore we have

$$\pi_{i+1}(\mathbf{a}|\mathbf{s}) = \frac{\mathbb{1}_{[Q_{\mathcal{B}}^{\pi_i}(\mathbf{s}, \mathbf{a}) \geq V^{\pi_i}(\mathbf{s})]} \mu_{\mathcal{B}}(\mathbf{a}|\mathbf{s})}{Z^{\pi_{i+1}}(\mathbf{s})}.$$

From the above equation, it is easy to see that $\forall \mathbf{a} \in$ supp $\pi_{i+1}(\cdot|\mathbf{s})$, $Q_{\mathcal{B}}^{\pi_i}(\mathbf{s}, \mathbf{a}) \geq V^{\pi_i}(\mathbf{s})$. Therefore $\sum_{\mathbf{a}} \pi_{i+1}(\mathbf{a}|\mathbf{s}) Q_{\mathcal{B}}^{\pi_i}(\mathbf{s}, \mathbf{a}) \geq V^{\pi_i}(\mathbf{s}) = \sum_{\mathbf{a}} \pi_i(\mathbf{a}|\mathbf{s}) Q_{\mathcal{B}}^{\pi_i}(\mathbf{s}, \mathbf{a}).$

$$Q_{\mathcal{B}}^{\pi_i}(\mathbf{s}, \mathbf{a})$$
$$= \mathbb{E}\left[ r(\mathbf{s}_t, \mathbf{a}_t) + \gamma \sum \pi_i(\mathbf{a}_{t+1}|\mathbf{s}_{t+1}) Q_{\mathcal{B}}^{\pi_i}(\mathbf{s}_t, \mathbf{a}_t) \Big| \mathbf{s}_t = \mathbf{s}, \mathbf{a}_t = \mathbf{a} \right]$$
$$\leq \mathbb{E}\left[ r(\mathbf{s}_t, \mathbf{a}_t) + \gamma \sum \pi_{i+1}(\mathbf{a}_{t+1}|\mathbf{s}_{t+1}) Q_{\mathcal{B}}^{\pi_i}(\mathbf{s}_t, \mathbf{a}_t) \Big| \mathbf{s}_t = \mathbf{s}, \mathbf{a}_t = \mathbf{a} \right]$$
$$\cdots$$
$$\leq \mathbb{E}_{\pi_{i+1}}\left[ \sum_{k=0}^{\infty} \gamma^k r(\mathbf{s}_{t+k}, \mathbf{a}_{t+k}) \Big| \mathbf{s}_t = \mathbf{s}, \mathbf{a}_t = \mathbf{a} \right]$$
$$= Q_{\mathcal{B}}^{\pi_{i+1}}(\mathbf{s}, \mathbf{a}).$$

&emsp; □

**Proposition 3.** *(Policy improvement) When using the Tabular CRR exp objective,* $Q_{\mathcal{B}}^{\pi_{i+1}}(\mathbf{s}, \mathbf{a}) \geq Q_{\mathcal{B}}^{\pi_i}(\mathbf{s}, \mathbf{a}) \, \forall \mathbf{s}, \mathbf{a} .$

*Proof.* We have that $KL(\pi_i(\cdot|\mathbf{s})||\mu_\mathcal{B}(\cdot|\mathbf{s})) \leq \epsilon \; \forall \mathbf{s}$. Since

$$\pi^{i+1} = \arg\max_\pi \mathbb{E}_{\mathbf{s}\sim d_\mathcal{B}}\left[\sum_\mathbf{a} Q_\mathcal{B}^{\pi_i}(\mathbf{s},\mathbf{a})\pi(\mathbf{a}|\mathbf{s})\right] \text{ s.t. } KL(\pi(\cdot|\mathbf{s})||\mu_\mathcal{B}(\cdot|\mathbf{s})) \leq \epsilon \; \forall \mathbf{s},$$

we know that $\sum_\mathbf{a} Q_\mathcal{B}^{\pi_i}(\mathbf{s},\mathbf{a})\pi_{i+1}(\mathbf{a}|\mathbf{s}) \geq \sum_\mathbf{a} Q_\mathcal{B}^{\pi_i}(\mathbf{s},\mathbf{a})\pi_i(\mathbf{a}|\mathbf{s}) \; \forall \mathbf{s}$.

It is easy to show that

$$Q_\mathcal{B}^{\pi_i}(\mathbf{s},\mathbf{a})$$

$$= \mathbb{E}\left[r(\mathbf{s}_t,\mathbf{a}_t) + \gamma\sum \pi_i(\mathbf{a}_{t+1}|\mathbf{s}_{t+1})Q_\mathcal{B}^{\pi_i}(\mathbf{s}_t,\mathbf{a}_t)\Big|\mathbf{s}_t = \mathbf{s}, \mathbf{a}_t = \mathbf{a}\right]$$

$$\leq \mathbb{E}\left[r(\mathbf{s}_t,\mathbf{a}_t) + \gamma\sum \pi_{i+1}(\mathbf{a}_{t+1}|\mathbf{s}_{t+1})Q_\mathcal{B}^{\pi_i}(\mathbf{s}_t,\mathbf{a}_t)\Big|\mathbf{s}_t = \mathbf{s}, \mathbf{a}_t = \mathbf{a}\right]$$

$$\dots$$

$$\leq \mathbb{E}_{\pi_{i+1}}\left[\sum_{k=0}^\infty \gamma^k r(\mathbf{s}_{t+k},\mathbf{a}_{t+k})\Big|\mathbf{s}_t = \mathbf{s}, \mathbf{a}_t = \mathbf{a}\right]$$

$$= Q_\mathcal{B}^{\pi_{i+1}}(\mathbf{s},\mathbf{a}).$$

$\square$

Finally we show that the difference in $Q$ values for a given policy $\pi$ between the ground truth MDP and the empirical MDP reduces as $|\mathcal{B}|$ increases. This allows us to conclude that the policies learned in the empirical MDP $M_\mathcal{B}$ is also a sensible policy in the original MDP.

**Proposition 4.** *Consider*

$$\epsilon_{MDP}(\mathbf{s},\mathbf{a}) = Q^\pi(\mathbf{s},\mathbf{a}) - Q_\mathcal{B}^\pi(\mathbf{s},\mathbf{a}).$$

*Define sets defined as $S(\mathbf{s},\mathbf{a}) = \{(\bar{\mathbf{s}},\bar{\mathbf{a}},\bar{\mathbf{s}}') \in \mathcal{B} \mid \bar{\mathbf{s}} = \mathbf{s}, \bar{\mathbf{a}} = \mathbf{a}\}$. If as $|\mathcal{B}| \to \infty$, $S(\mathbf{s},\mathbf{a}) = \emptyset$ or $|S(\mathbf{s},\mathbf{a})| \to \infty$, and $\bar{\mathbf{s}}'$ is an i.i.d sample of $P(\cdot|\bar{\mathbf{a}},\bar{\mathbf{s}}) \; \forall \; (\bar{\mathbf{s}},\bar{\mathbf{a}},\bar{\mathbf{s}}') \in \mathcal{B}$, then*

$$as \; |\mathcal{B}| \to \infty, \quad \sup_{\substack{\mathbf{s}\in\text{supp } d_\mathcal{B} \\ \mathbf{a}\in\text{supp }\pi(\cdot|\mathbf{s})}} \epsilon_{MDP}(\mathbf{s},\mathbf{a}) \to 0.$$

*Proof.* From Lemma 1 in [10], we have

$$\epsilon_{MDP}(\mathbf{s},\mathbf{a})$$

$$= \sum_{\mathbf{s}'}\left(P_\mathcal{M}(\mathbf{s}'|\mathbf{s},\mathbf{a}) - P_\mathcal{B}(\mathbf{s}'|\mathbf{s},\mathbf{a})\right)\left(r + \gamma V_\mathcal{B}^\pi(\mathbf{s}')\right) +$$

$$\gamma\sum_{\mathbf{s}'}\left(P_\mathcal{M}(\mathbf{s}'|\mathbf{s},\mathbf{a}) - P_\mathcal{B}(\mathbf{s}'|\mathbf{s},\mathbf{a})\right)\sum_{\mathbf{a}'}\pi(\mathbf{a}'|\mathbf{s}')\epsilon_{MDP}(\mathbf{s}',\mathbf{a}') +$$

$$\gamma\sum_{\mathbf{s}'}P_\mathcal{B}(\mathbf{s}'|\mathbf{s},\mathbf{a})\sum_{\mathbf{a}'}\pi(\mathbf{a}'|\mathbf{s}')\epsilon_{MDP}(\mathbf{s}',\mathbf{a}')$$

Since $\mathcal{B}$ is coherent, we have

$$\epsilon_{MDP}(\mathbf{s},\mathbf{a})$$

$$\leq \sum_{\mathbf{s}'}\left(P_\mathcal{M}(\mathbf{s}'|\mathbf{s},\mathbf{a}) - P_\mathcal{B}(\mathbf{s}'|\mathbf{s},\mathbf{a})\right)\left(r + \gamma V^\pi(\mathbf{s}')\right) +$$

$$\gamma \sup_{\substack{\mathbf{s}\in\text{supp } d_\mathcal{B} \\ \mathbf{a}\in\text{supp }\pi(\cdot|\mathbf{s})}} \epsilon_{MDP}(\mathbf{s},\mathbf{a})$$

for all $\mathbf{s} \in \text{supp } d_\mathcal{B}, \mathbf{a} \in \text{supp }\pi(\cdot|\mathbf{s})$. Taking the supremum on both sides:

$$\sup_{\substack{\mathbf{s}\in\text{supp } d_\mathcal{B} \\ \mathbf{a}\in\text{supp }\pi(\cdot|\mathbf{s})}} \epsilon_{MDP}(\mathbf{s},\mathbf{a})$$

$$\leq \sup_{\substack{\mathbf{s}\in\text{supp } d_\mathcal{B} \\ \mathbf{a}\in\text{supp }\pi(\cdot|\mathbf{s})}}\sum_{\mathbf{s}'}\left(P_\mathcal{M}(\mathbf{s}'|\mathbf{s},\mathbf{a}) - P_\mathcal{B}(\mathbf{s}'|\mathbf{s},\mathbf{a})\right)\left(r + \gamma V^\pi(\mathbf{s}')\right)$$

$$+ \gamma \sup_{\substack{\mathbf{s}\in\text{supp } d_\mathcal{B} \\ \mathbf{a}\in\text{supp }\pi(\cdot|\mathbf{s})}} \epsilon_{MDP}(\mathbf{s}',\mathbf{a}').$$

Rearranging the terms:

$$\sup_{\substack{\mathbf{s}\in\text{supp}\,d_{\mathcal{B}} \\ \mathbf{a}\in\text{supp}\,\pi(\cdot|\mathbf{s})}} \epsilon_{MDP}(\mathbf{s},\mathbf{a})$$

$$\leq \sup_{\substack{\mathbf{s}\in\text{supp}\,d_{\mathcal{B}} \\ \mathbf{a}\in\text{supp}\,\pi(\cdot|\mathbf{s})}} \frac{1}{1-\gamma}\sum_{\mathbf{s}'}\left(P_{\mathcal{M}}(\mathbf{s}'|\mathbf{s},\mathbf{a}) - P_{\mathcal{B}}(\mathbf{s}'|\mathbf{s},\mathbf{a})\right)\left(r + \gamma V^{\pi}(\mathbf{s}')\right).$$

Let $R_{\max} = \frac{1}{1-\gamma}\max_{\mathbf{s},\mathbf{a}}|r(\mathbf{s},\mathbf{a})|$, we than have by Hölder's inequality

$$\sup_{\substack{\mathbf{s}\in\text{supp}\,d_{\mathcal{B}} \\ \mathbf{a}\in\text{supp}\,\pi(\cdot|\mathbf{s})}} \epsilon_{MDP}(\mathbf{s},\mathbf{a}) \leq \sup_{\substack{\mathbf{s}\in\text{supp}\,d_{\mathcal{B}} \\ \mathbf{a}\in\text{supp}\,\pi(\cdot|\mathbf{s})}} \frac{R_{\max}}{1-\gamma}\sum_{\mathbf{s}'}\left|P_{\mathcal{M}}(\mathbf{s}'|\mathbf{s},\mathbf{a}) - P_{\mathcal{B}}(\mathbf{s}'|\mathbf{s},\mathbf{a})\right|.$$

since $|S(\mathbf{s},\mathbf{a})| \to \infty$ as $|\mathcal{B}| \to \infty$ for all $\mathbf{s} \in \text{supp}\,d_{\mathcal{B}}$ $\mathbf{a} \in \text{supp}\,\pi(\cdot|\mathbf{s})$, we have $\sup_{\substack{\mathbf{s}\in\text{supp}\,d_{\mathcal{B}} \\ \mathbf{a}\in\text{supp}\,\pi(\cdot|\mathbf{s})}} \sum_{\mathbf{s}'}\left|P_{\mathcal{M}}(\mathbf{s}'|\mathbf{s},\mathbf{a}) - P_{\mathcal{B}}(\mathbf{s}'|\mathbf{s},\mathbf{a})\right| \to 0$ by Theorem 1 of [11]. $\qquad\square$

## A.2 Evaluation protocol

To compute the performance of each agent, as reported in the Tables 2, 3,5, 6 and 7, we adopt the following procedure. We run each agent with three independent seeds. Agent snapshots are made every 50000 learner steps. For every agent / environment / seed combination, we evaluate all its saved snapshots by running each for 300 episodes in the environment and record the mean episodic reward of the best snapshot as an agent's performance for the seed. The performance of an agent in an environment is thus calculated as the mean performance across its seeds, and error bars the standard deviation of the means.

## A.3 Effects of using K-step returns

In this section, we evaluate the effect of using K-step returns compared to our proposed method of estimating advantages. As discussed in Sec. 3 using K-step returns can hurt the agent's performance since the transitions stored in the dataset are likely to be generated by very different policies than the current one. As a result, K-step returns may not reflect the actual returns of the current policy being evaluated and therefore introduce a bias. To test this hypothesis, we evaluate CRR's (using the *binary max* rule) performance while estimating the advantage by

$$\sum_{i=0}^{k-1}\gamma^i r_{t+i} + \gamma^k\frac{1}{m}\sum_{j=1}^{m}Q_\theta(\mathbf{s}_{t+k},\tilde{\mathbf{a}}_{t+k}^j) - \frac{1}{m}\sum_{j=1}^{m}Q_\theta(\mathbf{s}_t,\tilde{\mathbf{a}}_t^j)$$

where $\tilde{\mathbf{a}}_{t+k}^j \sim \pi(\mathbf{s}_{t+k})$, and $\tilde{\mathbf{a}}_t^j \sim \pi(\mathbf{s}_t)$. This objective is similar to the ones used in [27, 7].

As shown in Fig. 5, when choosing $k = 5$, we indeed observe an degradation in performance. This confirms that, with a large enough $k$, K-step returns produce a bias that compromises learning.

## A.4 Hyper-parameters

In this section, we describe the hyper-parameters of algorithms used.

### A.4.1 BCQ

For BCQ, we mostly follow the original network architecture as well as hyper-parameter settings in our implementation. Please refer to [10] for more details. We only changed the batch size to be 1024 to stay compatible with CRR.

### A.4.2 CRR

For CRR training we use: target network update period 100; 21 atoms on a grid from 0 to 100 for the distributional critic. We use $\beta = 1$ for Eqn. (4) and for CWP resampling. We swept $\beta$

over $[0.1, 0.4, 0.7, 1, 1, 3]$ on few selected environments and did not find it to affect the results too much and therefore kept the natural setting of $\beta = 1$. For all experiments, we use 2 separate Adam optimizers [16] for actor and critic learning respectively. The learning rates are set to $10^{-4}$. For sequence datasets, we use the batch size of 128 and for non-sequence datasets 1024. To compute the advantage estimates (see Table 1), we set $m = 4$.

For the descriptions of the network architecture, please see Figure 3. On the manipulation suite, two camera observations are provided in each time step. In these environments, we duplicate the entire lower stack of the network before the concatenation with proprioceptive (action) features to accommodate the extra pixel observation.

### A.4.3 D4PG

For both the actor and critic, we use the Adam optimizer [16] with the learning rate of $1e - 4$; target network update period 100; 51 atoms on a grid from -150 to 150 for the distributional critic. We use D4PG implemented in Acme [14], following their network architectures and hyper-parameters. We used batch size 1024 for experiments on the manipulation suite and 256 for the rest of experiments. We experimented with CRR's network and hyper-paramters for D4PG, but the performance is inferior.

### A.4.4 BC

Our BC implementation shares of its hyper-parameters with CRR whenever applicable. Most hyper-parameters, however, do not apply to BC. Our BC implementation shares the same policy network structure with CRR.

### A.4.5 ABM

For ABM training we use mostly the same hyperparameters as CRR. ABM uses an additional prior policy network to train which we also use an Adam optimizer with learning rate $10^{-4}$. To compute the advantage estimates for ABM, we set use 20 samples as per the original paper. For ABM specific hyper-parameters, the please see Table 4.

Our ABM implementation shares the same network architecture with CRR with the exception of its policy head being Gaussian. For the descriptions of the network architecture, please see Figure 3.

### A.5 Value over-estimation

In section 4.3, we mention that the relative value of actions in the dataset compared to that of the policy tends to increase due to value overestimation (for actions present in the dataset). We hypothesize that this is due to the following effect. As the CRR policy gets better, the value of the

Figure 5: Evaluating the effect of K-step returns ($K = 5$ in this case). In offline RL, K-step returns could be detrimental as it introduces a bias. Here we see that using K-step returns hurts policy performance.

Table 4: ABM specific hyper-parameters.

| Hyper-parameters | values |
|---|---|
| Number of actions sampled per state | 20 |
| $\epsilon$ | 0.1 |
| $\epsilon_\mu$ | $5 \times 10^{-3}$ |
| $\epsilon_\Sigma$ | $1 \times 10^{-5}$ |

Figure 6: Here we compare the percentage of accepted actions under the *binary max* rule versus agents' performance measured by episodic return. Surprisingly, the two quantities are positively correlated suggesting that the values of some actions in the dataset are overestimated compared to that of the policy.

bootstrap target (in critic learning) increases. As a result, the Q values for actions in the dataset, even that of suboptimal actions, also increase. As a result some of the suboptimal actions would have values higher than the value of the current policy.

We demonstrate this effect by comparing the percentage of actions copied using the *binary max* rule versus the agents' performance measured in terms of episodic return. In these experiments, we evaluate the agents' performance between 200000 and 600000 learner steps. We do not include datapoints corresponding to fewer than 2000000 learner steps in this analysis since early in learning the critic may not be reliable. We, in addition, measure how the percentage of actions copied by the *binary max* rule in the same learner steps range.

Intuitively, as the agent's performance improves, we would expect the dataset to contain fewer actions that outperform the agent's policy. Thus, the fraction of actions included in the policy update in

Figure 7: CWP ablation on selected environments. In these environments, we find CWP to mostly help and when it does not, CWP also do not hurt policy performance.

Eqn. 3 should go down. We do observe the opposite (as shown in Figure 6), however: the number of actions that are included in the policy update *increases* as agent performance improves. This supports our theory that the value of some actions from the datasets are overestimated compared to the value of the policy.

Due to the relative overestimation of the datasets' actions' values, CRR could clone sub-optimal actions leading to degradation in performance. The *binary max* rule, by being optimistic about the policy's state-value, copies fewer sub-optimal actions and thereby increases CRR's performance on some datasets.

Figure 8: Control suite policy noise ablation.

## A.6    Effect of noise when running the policy.

As mentioned in Section 4.2, we turn off the noise of component Gaussian distributions when evaluating policies. In Figure 8 and 9, we compare the effect of turning off the noise versus not. When it comes to environments of low and moderate action dimensionality, having no noise is beneficial, but often its effect is not significant. For examples, see results for *cartple swingup*, *walker stand*, *cheetah run* in Figure 8.

For some environments the effect of noise is pronounced. This is clearly visible in Fig. 9 for instance. For the humanoid environments in the locomotion suite, the task is terminated when the humanoid falls. This makes it hard to recover from mistakes and may put policies with a high-level of stochasticity at a disadvantage. Overall, turning off the noise rarely adversely affect the agents' performance.

Figure 9: Locomotion policy noise ablation.

## A.7 CWP Ablation

In this section we ablate the effect of CWP. Notice, when using CWP, we also turn off the component Gaussian distributions' noise as described in the previous section. As we use mixture of Gaussians policies, CWP essentially chooses between the mean actions of the component Gaussians.

For the full list of results, please refer to Tables 5, 6, and 7. In Figure 7, we pick a few representative environments to illustrate CWP's effect. As shown in the results, CWP generally helps with its effect especially pronounced when the policies' performance is less than ideal. When CWP does not help, it also does not lower policy performance. CWP, however, does require more computation. We therefore recommend CWP whenever it is not too expensive to execute.

## A.8 Full results table

Finally, we present all agents's performance in Tables 5, 6, and 7.

Table 5: Results on Deepmind Control suite. We divide the Deepmind control suite environments into two rough categories: easy (first 6) and hard (last 3).

| | BC | D4PG | RABM | BCQ | RCRR exp no-CWP | RCRR exp | RCRR binary no-CWP | RCRR binary | RCRR binary max no-CWP | RCRR binary max |
|---|---|---|---|---|---|---|---|---|---|---|
| Cartpole Swingup | $386 \pm 6$ | $855 \pm 13$ | $798 \pm 30$ | $444 \pm 15$ | $607 \pm 22$ | $664 \pm 22$ | $859 \pm 9$ | $\mathbf{860 \pm 7}$ | $853 \pm 7$ | $858 \pm 15$ |
| Finger Turn Hard | $261 \pm 39$ | $764 \pm 24$ | $566 \pm 25$ | $311 \pm 38$ | $620 \pm 39$ | $714 \pm 38$ | $714 \pm 32$ | $755 \pm 31$ | $805 \pm 23$ | $\mathbf{833 \pm 57}$ |
| Walker Stand | $386 \pm 6$ | $\mathbf{929 \pm 46}$ | $689 \pm 13$ | $501 \pm 5$ | $668 \pm 10$ | $797 \pm 30$ | $820 \pm 21$ | $881 \pm 13$ | $908 \pm 17$ | $929 \pm 10$ |
| Walker Walk | $417 \pm 33$ | $939 \pm 19$ | $846 \pm 15$ | $748 \pm 24$ | $798 \pm 2$ | $901 \pm 12$ | $920 \pm 18$ | $936 \pm 3$ | $949 \pm 1$ | $\mathbf{951 \pm 7}$ |
| Cheetah Run | $407 \pm 56$ | $308 \pm 121$ | $304 \pm 32$ | $368 \pm 129$ | $544 \pm 54$ | $\mathbf{577 \pm 79}$ | $459 \pm 30$ | $453 \pm 20$ | $394 \pm 42$ | $415 \pm 26$ |
| Fish Swim | $466 \pm 8$ | $281 \pm 77$ | $527 \pm 19$ | $473 \pm 36$ | $501 \pm 11$ | $517 \pm 21$ | $587 \pm 24$ | $585 \pm 23$ | $\mathbf{599 \pm 21}$ | $596 \pm 11$ |
| Manipulator Insert Ball | $385 \pm 12$ | $154 \pm 54$ | $409 \pm 4$ | $98 \pm 29$ | $583 \pm 16$ | $625 \pm 24$ | $640 \pm 26$ | $\mathbf{654 \pm 42}$ | $631 \pm 12$ | $636 \pm 43$ |
| Manipulator Insert Peg | $324 \pm 31$ | $71 \pm 2$ | $345 \pm 12$ | $194 \pm 117$ | $384 \pm 9$ | $\mathbf{387 \pm 36}$ | $373 \pm 34$ | $365 \pm 28$ | $337 \pm 23$ | $328 \pm 24$ |
| Humanoid Run | $382 \pm 2$ | $1 \pm 1$ | $302 \pm 6$ | $22 \pm 3$ | $586 \pm 6$ | $\mathbf{586 \pm 6}$ | $417 \pm 6$ | $412 \pm 10$ | $226 \pm 7$ | $226 \pm 11$ |

Table 6: Results on manipulation environments.

| | BC | D4PG | RABM | RCRR exp no-CWP | RCRR exp | RCRR binary no-CWP | RCRR binary | RCRR binary max no-CWP | RCRR binary max |
|---|---|---|---|---|---|---|---|---|---|
| box | $166 \pm 3$ | $9 \pm 9$ | $322 \pm 1$ | $340 \pm 4$ | $341 \pm 4$ | $343 \pm 5$ | $351 \pm 3$ | $351 \pm 4$ | $\mathbf{354.4 \pm 4.8}$ |
| insertion | $172 \pm 9$ | $0 \pm 0$ | $271 \pm 3$ | $308 \pm 7$ | $\mathbf{316.3 \pm 15.0}$ | $305 \pm 3$ | $312 \pm 11$ | $304 \pm 13$ | $309 \pm 7$ |
| slide | $91 \pm 9$ | $0 \pm 0$ | $262 \pm 4$ | $314 \pm 5$ | $322 \pm 11$ | $317 \pm 12$ | $\mathbf{325.7 \pm 9.4}$ | $321 \pm 3$ | $322 \pm 16$ |
| stack banana | $230 \pm 9$ | $0 \pm 0$ | $296 \pm 10$ | $320 \pm 10$ | $325 \pm 7$ | $328 \pm 7$ | $\mathbf{341.8 \pm 0.7}$ | $322 \pm 11$ | $328 \pm 15$ |

Table 7: Results on locomotion environments.

| | BC | D4PG | RABM | RCRR exp no-CWP | RCRR exp | RCRR binary no-CWP | RCRR binary | RCRR binary max no-CWP | RCRR binary max |
|---|---|---|---|---|---|---|---|---|---|
| Humanoid Corridor | $220 \pm 194$ | $4 \pm 4$ | $64 \pm 3$ | $902 \pm 38$ | $\mathbf{918 \pm 14}$ | $384 \pm 185$ | $484 \pm 97$ | $243 \pm 146$ | $245 \pm 180$ |
| Humanoid Gaps | $149 \pm 9$ | $5 \pm 3$ | $94 \pm 9$ | $537 \pm 55$ | $\mathbf{546 \pm 36}$ | $152 \pm 5$ | $149 \pm 89$ | $29 \pm 17$ | $33 \pm 21$ |
| Rodent Gaps | $463 \pm 137$ | $176 \pm 6$ | $420 \pm 70$ | $941 \pm 17$ | $\mathbf{957 \pm 19}$ | $485 \pm 72$ | $492 \pm 117$ | $368 \pm 36$ | $392 \pm 6$ |
| Humanoid Walls | $138 \pm 77$ | $2 \pm 1$ | $131 \pm 25$ | $371 \pm 17$ | $\mathbf{422 \pm 24}$ | $294 \pm 45$ | $289 \pm 72$ | $213 \pm 27$ | $232 \pm 24$ |
| Rodent Escape | $388 \pm 3$ | $24 \pm 14$ | $441 \pm 16$ | $420 \pm 14$ | $428 \pm 26$ | $451 \pm 37$ | $444 \pm 45$ | $\mathbf{501 \pm 29}$ | $499 \pm 40$ |
| Rodent Mazes | $343 \pm 48$ | $53 \pm 1$ | $\mathbf{478 \pm 7}$ | $471 \pm 5$ | $459 \pm 7$ | $463 \pm 14$ | $464 \pm 12$ | $460 \pm 18$ | $457 \pm 13$ |
| Rodent Two Tap | $325 \pm 60$ | $16 \pm 2$ | $598 \pm 2$ | $532 \pm 32$ | $543 \pm 32$ | $615 \pm 18$ | $\mathbf{615 \pm 19}$ | $584 \pm 17$ | $588 \pm 14$ |