[Reviews · NeurIPS 2020]

Review 1

Summary and Contributions: This paper proposes a simple yet effective method by filtering off-distribution actions in the domain of offline RL. The extensive experiments support the paper's claims about the proposed method's superior performances.

Strengths: - The method is well-motivated using simple examples. The basic idea is simple, yet the authors carry out the evaluation of the ideas well. - The authors show insightful connections with regularized policy iteration - Experiments are extensive. Overall it's a solid paper.

Weaknesses: - Maybe the authors need to explain more about the motivation behind the choice of equation (4) and its connection with equation (3). Although there is an analysis in section 4.3, one might raise a question: What might be other possible choices of the function f and why you don't choose them instead? - Although the authors have detailed the hyperparameters on the appendix, there are lots of other uncertainties in reimplementing this ideas, which might lead to problems of reproducibility. Unfortunately, there is no code submitted along with the paper. That said, in the Reproducibility section, the answer is Yes due to the scope limit is about the major results. - Throughout the paper, the authors consistently emphasize directly or indirectly that the idea of CRR is simple, yet its implementation seems not to be the case. For example, the network in Figure 3, while by no means using any new design of layers or connections, can hardly be considered simple. One might argue that: if the method is really good, why don't you just use simple, effective networks such as VGG or ResNet or simpler ones to prove its performances? Likewise, there is also a need to explain why you design the network as in Figure 3.

Correctness: The derivations look correct and the experiments are carefully and extensively undertaken.

Clarity: The paper is very well written and easy to follow.

Relation to Prior Work: This part is discussed in good detail.

Reproducibility: Yes

Additional Feedback: Post Rebuttal: After reviewing the responses from the authors, and other reviews, I would like to keep the score. Thank you for your effort in responding to my questions.


Review 2

Summary and Contributions: This paper proposes an offline RL method using an actor and a critic. The critic is trained using distributional Q-learning and the policy is trained using a weighted behavioral cloning loss. The weighting of each state action pair is determined as a function of the some advantage estimate using the critic. They consider both binary and soft advantage weighting as well as various advantage estimates.

Strengths: This paper has good experimental results and the method seems reasonable and straightforward. There is a is a detailed discussion of the similarities and differences with prior work.

Weaknesses: The authors discusses problems with bootstrapping but still uses it to train the Q function; they just assume that "Q is sufficiently accurate for (s, a) ∈ B". Moreover the choice of the weighting function seems somewhat heuristic and can have different performance depending on the task complexity and demonstration quality.

Correctness: The use of bootstrapping in the Q-learning seems odd. If the Q-function is not reliable, why would a weighting based on the Q function work better? If the Q-function is good, why cant it be used for improving the policy? How do we know if Q is sufficiently valid for (s, a) ∈ B? Perhaps the authors can include more discussions of these questions.

Clarity: Parts of the discussion in section 3.2 seem overly concise and hard to follow. I would have liked to see the main discussions of the method be more detailed and flushed out. The other sections of the paper are generally fine.

Relation to Prior Work: The paper has a good discussion on how it differs from prior works. It is also similar to many prior works and draws inspiration from them but is different in subtle ways.

Reproducibility: Yes

Additional Feedback: Update after author response: I would have liked to see the reasoning for the method's choices more flushed out in paper, and it would be good if the authors can include parts of their response in the paper itself. Generally the experimental results are good, the idea is straightforward, but parts of the paper can be made more detailed. I will update my score to marginally above the acceptance threshold


Review 3

Summary and Contributions: This paper studies the problem of offline reinforcement learning. The authors propose a novel offline RL algorithm using a form of critic-regularized regression. Empirical studies show that the algorithm achieves better performance on benchmark tasks.

Strengths: The problem of offline policy learning is important in both academic and industrial applications. The authors have a sound understanding of related works and propose novel methods to tackle key problems in offline RL. The topic is relevant to the NeurIPS community.

Weaknesses: Post rebuttal: Thanks for the authors response which clarifies most questions about the current work. I updated my score from 4 to 5 after rebuttal. My remaining concern is about the novelty part. At first glance the Algorithm 1 is like an "actor-critic" method with monotonic advantage weighting and distributional Q (with bootstrapped Q estimation)? In the introduction (line 30-32) and conclusion part (line 304-306) it seems that the point proposed is about filtered BC with advantage re-weighting. In rebuttal the point seems to be a simple and effective algorithm combined. I may suggest a major revision to restructure the work for a clear point of contribution. Minor: 1. The comparison with K-step/return-based method: The comparison in A.2 is indeed about the difference between A^{\pi_\theta} and A^{\mu_B}? Maybe a fair comparison near line 153 is a K-step/return-based estimation of A^{\pi_\theta}? There are methods can correct (to some extent) the advantage estimation from behavior policy \mu_B to the target policy \pi_\theta, e.g. Retrace and V-trace. 2. The name of A_mean and A_max may be misleading, personally I may suggest to define the truncation threshold in Eq.(3), instead of in advantage estimation. 3. And I also notice that the notation of f in Eq.(2) may need further revision. It seems that \pi will not be optimized within f(), and f is not dependent on action a if conditioned on Q_\theta(s,a). ---- I think the main contribution of this paper is a little ambiguous. I would be happier if the authors can describe this more explicitly. And I also have some detailed questions about correctness and novelty, which will be discussed in the following parts. 1. For the advantage estimation part, it seems that a constant depending on s_t will not affect the value of (2) (it cancels with the log pi part), so the difference between A_mean and A_max in Table 1 is unclear to me. 2. The convergence of the algorithm is not fully discussed. If Alg 1 converges, what would be the final Q_theta? It seems that Q will converge to Q_B instead of Q^{\pi_\theta}? With this estimation of Q, what would be the final \pi_\theta if it converges? A local optimal policy with respect to some loss function? 3. For the empirical performance, is there any possibility that the performance of MARWIL[35] BAIL[7] AWR[25] can also be compared? Since the algorithm have not been proved better theoretically, we expect stronger empirical results to demonstrate the power of the algorithm. Novelty: In Sec3 we see some ideas appeared in previous work, e.g. Eq(1) is about how to estimate Q, with distributional Q in [4], Eq(2) is about monotonic advantage term, similar to that of [35] etc. And for the estimation of advantage, the discussion of off-policy friendly methods like Retrace and V-trace is missing. Retrace: Safe and Efficient Off-Policy Reinforcement Learning V-trace: IMPALA: Scalable Distributed Deep-RL with Importance Weighted Actor-Learner Architectures

Correctness: The claims and method are mostly correct. Convergence property is not fully discussed.

Clarity: The paper is mostly clearly written. The rigorous definitions of notation (Q_\theta, \hat{A}) are missing.

Relation to Prior Work: The paper discussed prior related works. It would be better if the main contribution and difference can be discussed more explicitly.

Reproducibility: Yes

Additional Feedback:


Review 4

Summary and Contributions: This paper proposes an approach to batch RL. When training from pre-recorded trajectory, the technique only trains using the steps that are better than what the current policy wants to do. Examples where the trajectory is worse than the choice of the current policy are omitted from training.

Strengths: The idea is simple to understand. The idea seems like it has general applicability to most batch RL algorithms. This helps significance of the work to the field as it will likely be a useful tool for further research. Evaluation is performed across a wide range of environments and with modern SOTA comparisons.

Weaknesses: If I was unkind I might say that the simplicity of the technique prevents it from being particularly novel, and that the paper cannot bring much new knowledge to the field. It seems unlikely that many deeper insights remain to be gleaned based on this idea, which adversely affects the impact.

Correctness: As far as I can see the paper is technically correct. This is unsurprising given the simplicity of the idea.

Clarity: The idea is simple and is quite clearly articulated in the paper.

Relation to Prior Work: I am not an expert in batch RL but related work seems good with up to date references and comparisons

Reproducibility: Yes

Additional Feedback: Post-rebuttal: After reading the other reviews and rebuttal, I am happy to maintain my previous rating.

[Author Response · NeurIPS 2020]

We would like to thank the reviewers for your thoughtful feedback and comments which would undoubtedly make the
paper better. We will update our paper to reflect your comments, fix typos and include missing references. Here, we
aim to address the concerns of each reviewer.

**R4 & R5 – main contribution / insights:** Our goal was to devise an effective algorithm that works well on a diverse
set of offline RL problems. We consider this to be the main contribution of our paper: a simple algorithm which achieves
good results on a large variety of environments/dataset, including challenging high-dimensional, partially observed
problems from pixels. Although some components of our algorithm have been considered in related recent or concurrent
work, this has usually been the case in the context of more complicated formulations. A secondary contribution of our
work is thus a careful analysis and ablation that sheds light on the important attributes, and to identify a particularly
simple but effective combination. We will update the paper to make this more overt.

**R2 & R3 – motivation behind Eq. 3 and 4**: One intuition for Eq. 4 (as explained in the subsection titled regularized
policy iteration) works as follows. Through exponential weighting, we approximately regularize the policy by how
much it diverges from the behavioral policy as measured by KL. The regularization would constrain the policy to not
deviate too much from the support of the dataset thus ensuring effective offline learning. Eq. 4 is therefore chosen
because of its connection to KL divergence. Both Eq. 3 and 4 are motivated by the policy improvement theorem.
Whereas Eq. 3 seeks to improve the policy by choosing a better action to copy, Eq. 4 does this in a soft manner. We
chose Eq. 3 because of its direct link to the policy improvement theorem. There indeed are many other choices of
filtering functions and finding the best is an interesting direction of future research.

**R2 – reproducibility:** We have open-sourced the code for CRR on Github and the link will be made available.

**R2 – network architecture:** Due to the complexity of the locomotion datasets, we use relatively large networks. For
consistency reasons, we adopt the same network architecture for the control suite dataset though smaller networks
would suffice. We contend that other than being large in size, the networks used are not very unconventional in the
RL community. The convolutional nets are actually borrowed from Impala and ACME where we only modified the
activation functions. The residual MLPs are chosen for their expressive power.

**R3 – bootstrapping:** In the offline setting, the problem with bootstrapping arises when the current policy selects
actions not in the support of the dataset. When this happens, the bootstrapping target can be inaccurate due to the lack
of data. Since CRR's policies are learned by copying the actions of the dataset, the actions selected by CRR would
be close to that of the dataset thereby making the problem much less severe. The fact that CRR does better than BC
supports the claim the Q function is learned well. D4PG's poor performance (despite the same value learning rule)
suggests naive strategies in learning the policy is not sufficient since it is not constrained to the support of the dataset.

**R3 – weighting functions:** It is indeed the case that the two weighting functions produce different results. This finding
makes it clear that not all weighting functions are created equal and thus opens new doors to future research. As offline
RL algorithms cannot escape the existence of hyper-parameters, the choice of weighting functions is treated as one
additional hyper-parameter.

**R4 – advantage estimation:** We agree that adding constants which depend only on $s_t$ to $f$ in Eq. 2 (i.e. $f + g(s_t)$)
does not make a difference. However, $f$ is defined as a nonlinear transformation of the advantage estimates. $A_{max}$
underestimates the advantage where as $A_{mean}$ does not. Thus, using $A_{mean}$ vs. $A_{max}$ leads to different outcomes.

**R4 – convergence of Q:** Our algorithm performs standard policy evaluation to learn $Q$ for the current policy, effectively
as an inner loop (Eq. 1). Thus, if our algorithm converges to a pair $(\pi, Q)$, $Q$ will estimate the value of the policy, i.e.
$Q^\pi$ (restricted to the data in the batch and subject to the usual caveats of RL with nonlinear function approximation). In
the tabular setting, under some mild regularity conditions, CRR constitutes policy improvements in a restricted MDP
similar to that defined in BCQ [10]. Therefore each iteration of CRR will result in a better policy. Via similar arguments
for the convergence of policy iteration, $Q$ of CRR would converge to one corresponding to an optimal policy for the
restricted MDP. The theoretical results would be included in an updated version of the paper.

**R4 – additional baselines:** Thanks for raising the question. We would like to point out that our experiments already
include several baselines that are considered to be the state of the art at the moment of writing. In our experiments, we
also examine elements of the baselines and compare it to the equivalent bits in our method. For example, in Appendix
A.2 we evaluate the effect of K-step returns used by AWR and MARWIL.

**R4 – novelty:** Our main contribution is a simple and effective algorithm that identifies important ingredients of offline
learning algorithms (see response "Main contribution" above). We do not claim that the use of distributional Q learning
is a contribution per se. We will make this clearer in the text. CRR uses a different advantages estimate compared to
[35]. We include a toy example (in subsection titled "CRR vs. return-based methods.") that shows how the approach
of [35] can fail. We also show empirically (in appendix A.2) that our advantage estimation methods significantly
outperforms that adopted by [35].

[Meta-Review · NeurIPS 2020]

This paper proposes a simple yet effective method by filtering off-distribution actions in the domain of offline RL. During the review process, some concerns were raised regarding the novelty of the paper (the method seems to be simple yet very effective, and there are not many deep theory or significantly new ideas). Correspondingly, there have been a long list of discussions on the paper. As a result, the reviewers believe that this kind of work also has its value, especially for practical applications (offline RL). Some constructive feedback were also provided for the authors to consider when writing the final version of the paper. BTW, the rebuttal is quite good and changed the mind of some reviewers.